# Using Anomaly Feature Vectors for Detecting, Classifying and Warning of Outlier Adversarial Examples

Nelson Manohar-Alers [* 1]   Ryan Feng [* 1]   Sahib Singh [2]   Jiguo Song [2]   Atul Prakash [1]

## Abstract

We present DeClaW, a system for detecting, classifying, and warning of adversarial inputs presented to a classification neural network. In contrast to current state-of-the-art methods that, given an input, detect whether an input is clean or adversarial, we aim to also identify the types of adversarial attack (e.g., PGD, Carlini-Wagner or clean). To achieve this, we extract statistical profiles, which we term as *anomaly feature vectors*, from a set of latent features. Preliminary findings suggest that AFVs can help distinguish among several types of adversarial attacks (e.g., PGD versus Carlini-Wagner) with close to 93% accuracy on the CIFAR-10 dataset. The results open the door to using AFV-based methods for exploring not only adversarial attack detection but also classification of the attack type and then design of attack-specific mitigation strategies.

## 1. Introduction

While deep neural networks (DNNs) help provide image classification, object detection, and speech recognition in autonomous driving, medical, and other domains, they remain vulnerable to adversarial examples. In this paper, we focus on the detection problem – determining whether an input is adversarial and even attempting to determine its type (e.g., PGD vs. Carlini-Wagner), so that potentially in the future, attack-specific countermeasures can be taken.

Existing detection techniques include training a second-stage classifier (Grosse et al., 2017; Gong et al., 2017; Metzen et al., 2017; Aigrain and Detyniecki, 2019), detecting statistical properties (Bhagoji et al., 2017; Hendrycks and Gimpel, 2016; Li and Li, 2017; Crecchi et al., 2019; Pang

---
[*]Equal contribution  [1]Computer Science and Engineering Division, University of Michigan, Ann Arbor, MI [2]Ford Motor Company, Dearborn, MI. Correspondence to: Ryan Feng <rtfeng@umich.edu>, Atul Prakash <aprakash@umich.edu>.

*Accepted by the ICML 2021 workshop on A Blessing in Disguise: The Prospects and Perils of Adversarial Machine Learning.* Copyright 2021 by the author(s).

et al., 2017), and running statistical tests (Grosse et al., 2017; Feinman et al., 2017; Roth et al., 2019). In contrast to our work, these approaches are focused on detection of an attack rather than determining the attack type. It is an open question as to whether attacks can be distinguished sufficiently. If the answer is yes, that may help provide more insight into how attacks differ, leading to more robust classifiers, or attack-specific mitigation strategies.

To explore the question, our hypothesis is that it is possible to do a fine-grained distributional characterization of latent variables at an intermediate hidden layer across natural input samples. Our hope is that adversarial samples are outliers with respect to the that characterization in different ways. Unfortunately, finding differences between natural and different types of adversarial inputs is not straightforward. Fig. 1 shows histograms of (standarized) layer input values for a NATURAL/CLEAN sample as well as for two (first-stage) white-box attacks (Fast Gradient Sign (FGM) and $L_\infty$ Projected Gradient Descent (PGD) with 20 steps and $\epsilon = 8/255$. over a CIFAR10 pretrained model. Each histogram shows frequency counts for z-scores of observed values. While these distributions exhibit some differences, the differences are minute, making discrimination among them challenging.

We propose an approach called DeCLaW (Detecting, Classifying, and Warning of adversarial examples) that aims to overcome this challenge by constructing a set of anomaly features, in the form of an *anomaly feature vector* (AFV), from an input image that better distinguishes among clean and adversarial images as well as several of the common attack types. The key insight is to collect and look at statistical values in outlier areas in Figure 1 (shown with a magnifying glass) and try to characterize their distributions for different types of attacks, generating a collection of anomaly features that form the input image's AFV. We find that, given a model to be protected, each anomaly feature in the vector induces some separation between different types of attacks from clean examples. While overlaps exist when looking at one anomaly feature, across multiple anomaly features, we find good success in distinguishing among attack types.

DeClaW uses a second stage neural network that is trained on AFV values rather than latent features. Given an image,

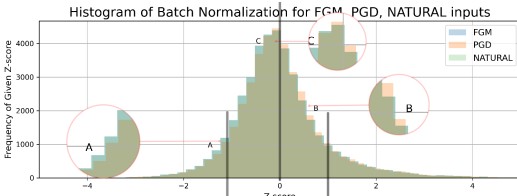

Figure 1: Histograms of (standarized) layer input values for one NATURAL sample and a adversarial versions from FGM and PGD20 of the same CIFAR10 sample. Attack perturbations tend to be small, but do induce potential differences when carefully analyzed.

its AFV is computed from the hidden layer values in the target classifier and then fed to the second stage neural network for analysis of the attack type. As a detector, using 11 different popular attack methods from the IBM Adversarial Robustness Toolbox (ART), DeClaW achieves 96% accuracy for CIFAR10 on adversarial examples and 93% accuracy on clean examples.

Notably, AFV size is small compared to input pixel values or the latent variables, resulting in a simpler second stage classifier. E.g., the current implementation of DeClaW uses only 176 anomaly detection features in contrast to approximately 16K latent features used by Metzen et al.'s adversarial example detector [18].

As an attack-type classifier, after grouping similar attacks into clusters, DeClaW achieves an overall 93.5% classification accuracy for CIFAR10. To our knowledge, we are the first to produce good attack classification. For example, whereas the work in (Metzen et al., 2017) reports detection rate on three attack methods (i.e., FGM, DeepFool, BIM), it does not discriminate between those underlying attacks. DeClaW is able to discriminate among these attacks with a high success rate.

## 2. Our Approach

DeClaW is both an attack labeling classifier as well as a binary attack detector. The fundamental idea is to augment a pretrained model with a sampling hook at a chosen layer to capture unperturbed data that can be analyzed and statistically characterized. That is then used to generate what we term as an anomaly feature vector when presented a perturbed input of a given type (e.g., adversarial attack generated by PGD). DeClaW's AFV generation sampling hook as shown in Fig. 2. The AFVs from the perturbed inputs as well as natural inputs of different classes are used to train a second stage neural network (Fig. 3) that can help distinguish among perturbed inputs and clean inputs and also help identify the type of perturbed input (e.g., a PGD attack versus a DeepFool attack). As we discuss later, some attack types may be difficult to distinguish reliably. When

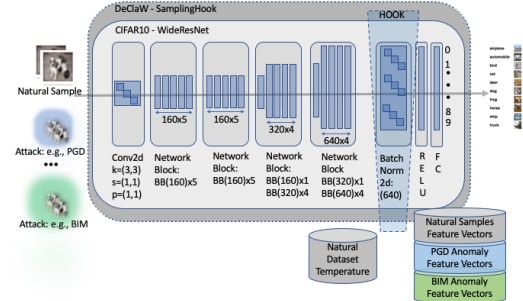

Figure 2: DeClaW subclassing extends sampling hook into the target pretrained model at a batch normalization layer that collects latent feature statistics on the dataset. Another pass generates AFVs for natural and adversarial samples.

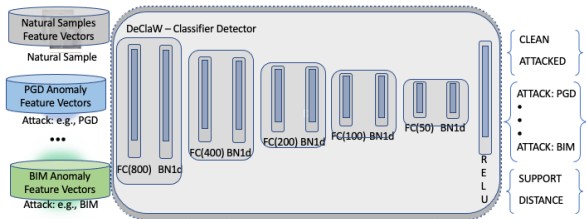

Figure 3: DeClaW's second stage classifier takes an image's AFV as input and outputs attack type (or clean). Dropout layers are used after each hidden layer for regularization.

that is the case, those attack types are clustered together using a clustering algorithm and treated as one composite class. Below, we describe the key concepts of the scheme.

One advantage of our approach is that the the size of AFV is designed to be independent of the latent feature vector size and is typically orders of magnitude smaller than the typical size of the feature vector from which it is derived. Thus, training and classification of an input using the second stage classifier of DeClaW is efficient. We elaborate on this in Section 3 (Experiments).

### 2.1. Anomaly Feature Vectors

An anomaly detection defense can be conceptualized as testing whether a sample is anomalous or not by checking whether statistics for the given sample are within the expected limits of the non-anomalous (natural) inputs. However, because the goal of sophisticated attacks is to resemble as much as possible a natural input, a number of coarse-grain $A/B$ statistical tests simply lack the resolution to reliably discern anomalies. We rely on various histogram comparative techniques, subsampling projections, aggregation predicates and distance metrics to account for shape and area differences between the histogram of expected layer inputs and the histogram for observed layer inputs associated with an evaluated input image.

DeClaW's neural network trains on a small and compact number of anomaly-detection distribution characterization features as opposed to on pixels or their derivative latent variables. The basic DeClaW second stage neural network uses 176 anomaly detection features – a significantly smaller number than the number of features used by Metzen et al.'s adversarial example detector (Metzen et al., 2017). In Metzen et al.'s detector, the preferred position reported is at stage $AD(2)$, the second residual block, where the inputs have a shape of $32 \times 32 \times 16 \approx 16K$ which is then fed to the detector convolution network. Moreover, the number of AFV features is independent of the input size since AFV features are essentially distributional characterization features. DeClaW trains fast, converging in few epochs.

Fig. 4 depicts three histogram plots corresponding to different DeClaW anomaly features evaluated across the same set of 5000 samples. On each plot, the blue shade represents the distribution of feature values for natural samples — each other color represents the distribution of feature values for the application of a given attack method to the same 5000 samples. Each anomaly feature is generated by applying a subsampling projection that zooms in a particular range of standardized scores (z-scores) and then by collecting aggregation metrics over the selected values (e.g., frequency counts of the selected z-scores). Each plot illustrates the discriminating power that such region-based aggregation feature induce. For example, the first plot shows a histogram for the values taken by an AFV feature that counts how many standardized scores are observed with value less than a minimum tolerance (i.e., the LEFT tail of the histogram).

We rely on four subsampling projections or regions: the LEFT tail (that is, low z-scores), the CENTER (that is, non-anomalous z-scores), the RIGHT tail (that is, high z-scores), and the OVERALL histogram – and ratios of these. Different aggregation predicates (e.g., count, sum, mean, ratio, etc) are used to aggregate over the subsampling projection.

A total of 12 different shades are shown corresponding to 11 attack methods that our second stage detector seeks to detect – plus the natural/clean samples shown in a blue shade.

Intuitively, whereas the detection goal is to separate the blue shade from other shades, the classification goal is to find combinations of features that allow to separate each shade from the others. Some features (e.g., Z.SUM()) induce linear separatability between natural samples and attack classes — hence, DeClaW's high detection accuracy and low FPR as well as FNR — whereas other features induce partial separatability between attack classes (hence DeClaW's high classification accuracy). For example, the above-mentioned Z.SUM() feature measures the sum of absolute standardized scores of layer-input values observed for a given sample and empirically discriminates well among AFVs from CLEAN/NATURAL (i.e. blue shade) from AFVs from any

adversarially perturbed samples (i.e., all others).

## 3. Experiments

We examine DeClaW classification performance with respect to one dozen different classes: one class 0 for Natural samples and 11 attack classes $A_1 \cdots A_{11}$ for attack-perturbed samples obtained from the corresponding application of 11 different evasion methods. Each of attack class $A_1 \cdots A_{11}$ was generated by applying the attack function of the IBM Adversarial Toolkit over the same subset of natural samples. Default parameters were used in the attacks. The Appendix lists the attack classes. Table 1. We also examine DeClaW detection performance with respect to the Class 0 above for Natural samples and the agglomeration of the above 11 classes aboves into one class representing attack-perturbed samples by any type of attack method. For most of the results in this paper, unless noted otherwise as in the case of clustering of attack classes that are too similar to discriminate, we focused on 11 different attack classes plus 1 class for the natural samples.

For the CIFAR-10 dataset, we used all the 60000 natural samples to train and test the second stage classifier. Using a subset of natural CIFAR-10 samples $C_6$, we first inferred the reference temperature of the dataset of the natural samples. For each attack class $A_k$, we collected the resulting anomaly-detection feature vector produced by our hook for the same $C_6$ natural samples. In total, we generated a total of 60000 anomaly detection feature vectors from the CIFAR-10 natural samples and a total of 102000 attack samples across the 11 attack classes. About $7.3\% = \frac{8000}{110000}$ of attacks failed to be successfully computed within our time bounds and we omitted those from consideration. We used a 70% to 30% split between train and test. More details of experimental setup are in the Appendix.

We use a 32-layer Wide Residual Network (Zagoruyko and Komodakis, 2016) as the classifier for CIFAR-10, with a sampling hook placed before the batch normalization stage of the model's pipeline (details in the Appendix). As a detector, for CIFAR-10, this network's accuracy on clean test data is 93% and 96% on adversarial data (see Fig. 5). This compares well to state-of-the-art baselines. For instance, Metzen et al. report 91.3% accuracy on clean data and 77% to 100% on adversarial data that used same permitted distortion during attacks as the detector was trained on (Metzen et al., 2017). Aigran et al. achieve only 55.7% false-positive rate when evaluated against FGSM attacks with a 95% true positive rate (Aigrain and Detyniecki, 2019).

Moreover, DeClaW (see Fig. 6) achieves high classification accuracy among attack methods, with similar attacks clustered. For instance, on adversarial versions of test data, it had almost perfect ability to distinguish among PGD and

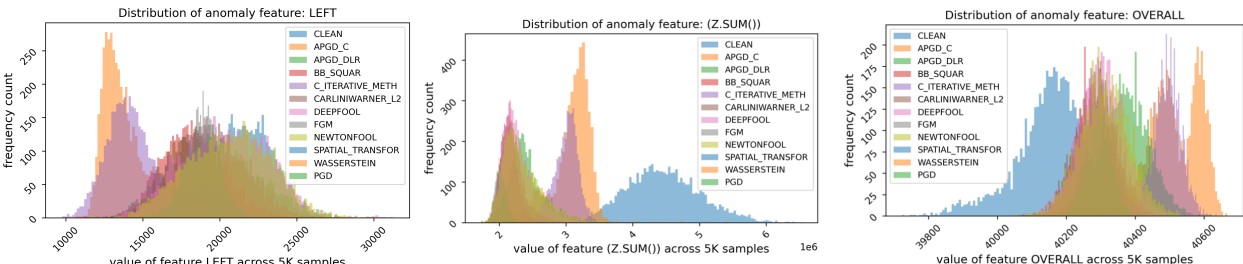

Figure 4: Each anomaly feature helps provide partial discrimination among attacks and clean examples. When used in combination, they can help towards classifying attack types and whether an example is clean.

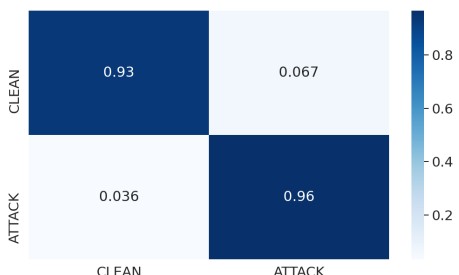

Figure 5: Confusion matrix for the detection of adversarial examples where CLEAN bit is set only if no ATTACK class is detected and ATTACK bit is set otherwise.

Carlini-Wagner attack images and close to 90% accuracy in distinguishing PGD attacks from non-PGD attacks. F1-scores, which can be computed from the confusion matrix, ranged from 90%-99% for the class groups shown. To our knowledge, we are the first to report such classification accuracy levels across different attack methods.

## 4. Limitations

Our approach shows promise on CIFAR-10 dataset in distinguishing among several types of attacks. But the approach should be tried on additional datasets to assess generalizability. Preliminary results on CIFAR-100 dataset are promising and given in the Appendix under Additional Results.

The approach also needs to be evaluated against adversarial attacks where the adversary has knowledge of the DeClaW pipeline. DeClaw pipeline, as it stands, is also not adversarially trained. We do note that in corporate environments, the DeClaW pipeline need not be made visible to an adversary, nor its results need to be directly visible, since it is primarily used for error detection and error classification rather than as the mainstream classifier (e.g., YouTube or Facebook could use an attack detector/classifier inspired by DeClaW on the backend for analysis without exposing that to end-users.) Furthermore, DeClaW pipeline could be adversarially trained. We note that for an adversary to

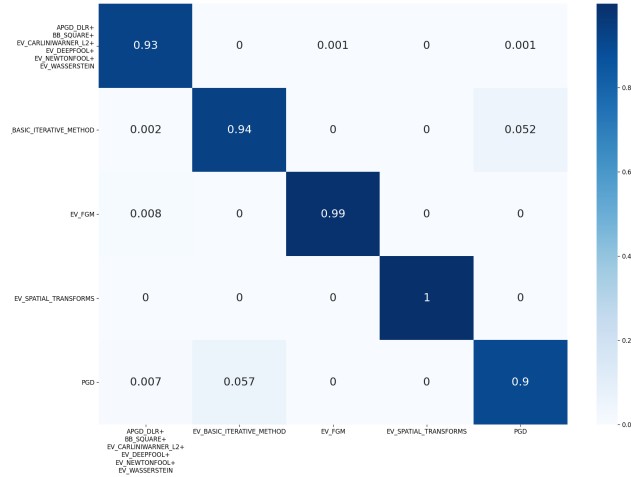

Figure 6: Declaw's classification accuracy among attacks after attack methods are clustered based on large FPR and FNR. Appendix (Fig. 9) is an expanded figure.

recreate the DeClaW pipeline, they require both the training data and whitebox access to the target model so as to create AFVs – a higher threshold than just whitebox access.

## 5. Conclusion

We present DeClaW, a system to detect, classify and warn of adversarial examples. We achieve high detection accuracy on CIFAR-10 (96% accuracy on adversarial data and 93% on clean data) while being the first to our knowledge to classify the type of attack. We are able to distinguish among many of the 11 different attacks on CIFAR-10 with F1-scores ranging from 90%-99%.

## Acknowledgment

We acknowledge research support from Ford and University of Michigan. This research is partially based upon work supported by the National Science Foundation under grant numbers 1646392 and 2939445 and DARPA award 885000.

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

# A. Appendix

## A.1. Attack Classes

Table 1 shows the attack classes that we trained and evaluated DeClaw on.

## A.2. Experimental Setup

For the CIFAR-10 dataset, we used all the 60000 natural samples to train and test the second stage classifier. Using a subset of natural CIFAR-10 samples $C_6$, we first inferred the reference temperature of the dataset of the natural samples.[1] For each attack class $A_k$, we collected the resulting anomaly-detection feature vector produced by our hook for the same $C_6$ natural samples. In total, we generated a total of 60000 anomaly detection feature vectors from the CIFAR-10 natural samples and a total of 102000 attack samples across the 11 attack classes.[2] We used a 70% to 30% split between train and test. – resulting in second stage datasets as follows:

- training with 42000 and testing with 18000 anomaly detection feature vectors for natural class $C$ and

- training with 72200 and testing with 29790 anomaly detection feature vectors across attacked classes $A_1 \ldots A_{11}$.

We performed the experiments using 12 classes:

- one class $C$ of anomaly detection feature vectors from natural samples (divided into subsets $C_1 \ldots C_6$, each of 10000 samples).

- and 11 classes $A_k$, each resulting from the application of a given attack class $A_k$ over the same (10000 sample size) subset $C_6$ of the natural samples.

DeClaW provides user control of various basic hyperparameters. These parameters are shown in Table 2.

We use a 32-layer Wide Residual Network (Zagoruyko and Komodakis, 2016) as the classifier for CIFAR-10. The network has been trained for 80 epochs with stochastic gradient descent and momentum on 10000 data points from the train set. The momentum term was set to 0.9 and the initial learning rate was set to 0.1, reduced to 0.01 after 50 epochs, and further reduced to 0.1 after 70 epochs. After each epoch, the network's performance on the validation data (the remaining 5000 data points from the train set) was

---

[1]In the Future Work section, we identify issues with generating a reference temperature of a dataset such as the presence of concept drift or multimodality.

[2]About 7.3% $= \frac{8000}{110000}$ of attacks failed to be successfully computed within time bounds.

determined. The network with maximal performance on the validation data was used in the subsequent experiments (with all tunable weights being fixed).

We attach an adversary detection subnetwork (called "detector" below) to the ResNet. The detector is a convolutional neural network using batch normalization (Ioffe and Szegedy, 2015) and rectified linear units. At this time, we focused on placing our sampling hook into the batch normalization stage of the original model's (e.g., CIFAR-10) pipeline.

## A.3. Clustering of similar attack types

DeClaW learns a statistical envelope for each attack class. However, as some attacks are more sophisticated derivative variants of other attacks, the resulting envelope may be similar.

Attacks are essentially perturbations optimized to reduce visible artifacts against the natural inputs. It is therefore to be expected that given the set of selected distributional features selected to identify those perturbations, some of these attacks would produce distributional profiles too close to each other to clearly differentiate between them. The result is that macro-averaged metrics across such confounded classes introduces losses in recall, precision, and F1 values that distort the perceived quality of the detector and classifier.

Higher detection and classification accuracy can be obtained by treating such clustered classes as one meta attack class. For example, we found that a set of twenty attack classes clustered into ten different similarity clusters of different sizes. Similarly, in this paper, our twelve attack classification classes were found to cluster into just the six different classification clusters identified in Table 1.

Our proposed solution is to cluster similar attacks. To do so, we use a transitive closure with respect to ranked FPR and FNR (being above a given tolerance (e.g., 10%), currently favoring FNR when conflicts arise. For example, given a normalized confusion matrix, the decision whether to cluster to attack classes $A_i, A_j$ is solely decided on the $2x2$ pairwise confusion matrix between those two attack classes — i.e.,

$$M(A_i, A_j) = \begin{pmatrix} acc(A_i) & fnr(A_i,A_j) \\ fpr(A_i,A_j) & acc(A_j) \end{pmatrix}.$$

Now, we threshold the above matrix, for $fpr$ and $fnr$ values greater than some threshold $t \geq 0.2$, we focus solely on large attack classification error. For example, given $M(A_i, A_j) = \begin{pmatrix} 1 & a \\ b & 1 \end{pmatrix}$ where $a, b$ could either be 0 or 1. If any is 1, attack classes $a, b$ are clustered as one. As a convention, we assign the parent class as the one with the lowest classification label. The same principle applies for larger

| Class Name | Class Num | Cluster | Samples | Class Description | Ref |
|---|---|---|---|---|---|
| CLEAN | 0 | 0 | 60k | Natural samples | (Krizhevsky et al., 2009) |
| $APGD_{ce}$ | 1 | 1 | 10k | Auto-PGD (cross entropy) | (Croce and Hein, 2020) |
| $APGD_{dlr}$ | 2 | 2 | 10k | Auto-PGD (logits ratio) | (Croce and Hein, 2020) |
| $BB_{square}$ | 3 | 2 | 10k | Square attack | (Andriushchenko et al., 2020) |
| $BIM$ | 4 | 3 | 10k | Basic Iterative Method | (Kurakin et al., 2017) |
| $CW_{l2}$ | 5 | 2 | 10k | Carlini Wagner l2 | (Carlini and Wagner, 2017) |
| $Deep$ | 6 | 2 | 10k | DeepFool | (Moosavi-Dezfooli et al., 2016) |
| $FGM$ | 7 | 4 | 10k | Fast Gradient Method | (Carlini and Wagner, 2017) |
| $Newton$ | 8 | 2 | 10k | Newton Fool | (Jang et al., 2017) |
| $Spatial$ | 9 | 5 | 10k | Spatial Transforms | (Engstrom et al., 2019) |
| $Wasserst.$ | 10 | 2 | 10k | Wasserstein Attack | (Wong et al., 2020) |
| $PGD$ | 11 | 6 | 10k | Projected Grad. Descent | (Madry et al., 2019) |

Table 1: Attack classes used in the baseline experiment. In the non-clustered attack mode, 12 classes were used. In the clustered attack classification mode, those 12 classes were grouped into 6 cluster groups based on a transitive closure against either FPR or FNR $> 25\%$. Attack classes that are too difficult to discriminate between each other (and thus have high inter-class FPR or FNR) are then assigned to the same cluster. This clustering-by-FPR/FNR threshold technique is discussed in Section A.3.

| Parameter | Values | Default | Description |
|---|---|---|---|
| batch_size | $1 \cdots 10000$ | 2500 | batch size for training |
| sgd_mode | True/False | 0 | SGD if True else Adam |
| learning_rate | real number | 1.00 | learning rate |
| num_epochs | $1 \cdots$ | 20 | num. of epochs to train |

Table 2: DeClaW's command line hyper-parameters.

| LR | AUG | C0 F1 | AVG F1 | DET MuACC | CLF MuACC | N |
|---|---|---|---|---|---|---|
| 0.01 | 0 | 0.909 | 0.929 | 0.931 | 0.883 | 51 |
| 0.01 | 1 | 0.916 | 0.93 | 0.936 | 0.897 | 86 |
| 1.0 | 0 | 0.935 | 0.95 | 0.951 | 0.929 | 255 |
| 1.0 | 1 | 0.939 | 0.95 | 0.952 | 0.932 | 243 |

Table 3: Best performing parameters and their resulting accuracy metrics (F1 values for natural and across classes as well as average observed accuracy for detection and classification mode over a grid parameter evaluation. Attack classes were grouped into clusters based on a proxy to similarity, the corresponding $FNR$ and $FPR$ rates between two attack classes. A transitive closure reduced a dozen attack classes into six. Using this criteria, all models are in a plateau, whose maximal point is $94.2\%$ for classification $Acc_{C+A_k}$ and $90.8\%$ for detection $Acc)C|A$.

number of classes, except that a transitive closure over large classification error is used to drive the merge. As the number of classes is small, this process is done by hand, however, a union-join() or dfs() algorithm can be used to implement the transitive closure for the case when the number of classes to be defended is extremely large or it is desirable that such procedure is automated. Table 3 compares detection and classification accuracy metrics when using the clustered (6 attack clusters) mode. After clustering attacks that were found to be difficult to distinguish based on AFVs, reducing the number of classification labels from 12 to 6, the average classification accuracy increased by $21\%$ while average detection accuracy remained the same.

## A.4. Feature Generation Strategy

Rather than training our second stage network with pixels as features, we trained the second stage classifier using anomaly-detection features. For each input sample $x_i$, we examined the resulting input tensor $l_i$ presented to the sampling hook layer. That is, $l_i$ represents the layer inputs to the final batch normalization layer shown in Table 2.

Given that our CIFAR-10 input tensor $l_i$ to the batch normalization layer contains 640 channels of $8 \times 8$ features, our sampling hook observes flattened vectors $v_i$ containing 40960 values. We compare statistical features of these vectors derived from subsampling projections from it and comparing the outlier distribution against normative mean, variance, and count references. By examining the observed features location relative to channel-wise means $\mu_c$ and standard deviations $\sigma_c$, we identify outliers that contribute to an image being clean or adversarial, and if it's adversarial, which type of attack it is.

In total, we consider eight categories of features: region-based, extrema-based, histogram counts, histogram bin comparisons, probability tests, Wasserstein distances, PCA/LDA dimensionality reduction features, and nearest neighbor features.

### A.5. Basic Building Blocks for Anomaly Features

The input tensor values $l_i$ have a shape of $[1, 640, 8, 8]$ — meaning each sample has 640 channels of 64 values each. By considering each tensor $l_i$ as a vector in $R^{40960}$ of random variables, we estimate means and variances within each of the 640 channels as well as across all the 640 channels of $l_i$. For convenience, let $L = len(l_i)$, – that is, $L = 640 \times 8 \times 8 = 40960$ for CIFAR10.

We first estimate the baseline channel-wise means and standard deviations, notated as $\mu_c$ and $\sigma_c$ for a channel $c$. For a given input tensor $l_i$ and its values in channel $c$ notated as $l_{i,c}$, we define its mean $\mu_c(l_i) = \frac{\sum l_{i,c}}{L}$ and standard deviation $\sigma_c(l_i) = \sqrt{\frac{\sum (l_{i,c} - \mu_c(l_i))^2}{L}}$. Then, similar to the process in the original batch normalization paper (Ioffe and Szegedy, 2015), we compute running means and standard variances for each channel with an exponential smoother to compute our baseline means. For a channel $c$, let $\mu_{base,c}$ refer to the baseline mean and let $\sigma_{base,c}$ refer to the baseline standard deviation.

We hypothesize that using aggregation schemes to accumulate outliers would yield stronger anomaly detection signals. To this end, we set $lo$ and $hi$ watermarks with respect to distribution of values in a channel $c$. Trivially, these watermarks are (as is commonly done) implemented as $\sigma$-based control limits – specifically, $lo_c = \mu_{base,c} - \sigma_{base,c}$ and $hi_c = \mu_{base,c} + \sigma_{base,c}$.

Given $lo$ and $hi$ control limits or watermarks for each channel, we create three indicator vectors $S_{lo}, S_{mi}, S_{hi}$ for all $l_i$ that places each feature into one of those three disjoint sets. Let $v_i$ refer to a flattened vector of $l_i$ and $z_i$ refer to a z-scored version of $v_i$. For any given index $k$, and channel $c$ corresponding to the channel that $v_i[k]$ belongs to, the value of $S_{lo}[k] = 1$ if $v_i[k] \leq \mu_{base,c} - \sigma_{base,c}$ and is 0 otherwise. Likewise, the value of $S_{hi}[k] = 1$ if

$v_i[k] \geq \mu_{base,c} + \sigma_{base,c}$ and is 0 otherwise. The value of $S_{mi}[k] = 1$ if $\mu_{base,c} - \sigma_{base,c} < v_i[k] < \mu_{base,c} + \sigma_{base,c}$ and is 0 otherwise. Finally, we let an overall outlier vector $S_{ov}$ be a vector where its value is 1 where either $S_{lo}$ or $S_{hi}$ is 1 and 0 otherwise.

Using the above 4 indicator vectors, we generate 2 basic aggregation-based anomaly features for each vector $S_x$:

- $C(S_x) = \sum\limits_{\forall k \in [1,L]} S_x[k]$

- $Z(S_x) = \sum\limits_{\forall k \in [1,L]} S_x[k] \cdot z_i[k]$

For any input tensor $l_i$, these 8 scores are computed and then used as building blocks to generate features used by DeClaW.

Some features also compare against values from a reference dataset of representative samples. Let $D_N$ refer to such a dataset, formed from 10k randomly chosen values from the natural samples in the training set.

### A.6. Region-based Features

With the building block and aggregatation features described in Section A.5, we specify region-based features. These region-based features count and measure the signal of outliers in the left and right regions of the distribution curve. There are 25 features derived from the $C(S_x)$ and $Z(S_x)$ building blocks as described in Table 4.

### A.7. Extreme Value Features

The above region-based features in Section A.6 deal with central and fat portions of the tails of the distribution of $l_i$ values. Next, we develop features that isolate extrema tail values of the distribution of $l_i$ values. Let $R = 25$ be the number of region-based features above. For each feature $f_r, \forall r \in [1, 25]$, we note individual feature extrema values. For a given $f_r$, let $\phi_{lo}(f_r)$ and $\phi_{hi}(f_r)$ be extrema percentile thresholds (e.g., $lo = 10\%$ and $hi = 90\%$). That is, 10% of feature values $f_r(l_i^N)$ from the population of samples $l_i^N$ from $D_N$ are less than $\phi_{lo=10\%}(f_r)$. Similarly, 90% of feature values $f_r(l_i^N)$ are less than $\phi_{hi=90\%}(f_r)$. Finally, we define indicator flags $e_r(l_i)$ that identify, for $l_i$, whether or not the feature value $f_r(l_i)$ belongs to the normative population of $f_r$ values.

$$e_r(l_i) = \begin{cases} 1, & \text{if } f_r(l_i) \geq \phi_{hi=90\%}(f_r) \\ 1, & \text{if } f_r(l_i) \leq \phi_{lo=10\%}(f_r) \\ 0, & \text{otherwise} \end{cases}$$

$\forall r \in [1, R]$.

For each feature value $f_r(l_i)$ of every layer input $l_i$, values within the range $[(\phi_{lo}(f_r), \phi_{hi}(f_r))$ will be considered nor-

| Feature name | Region-based feature specification |
|---|---|
| error density | $f_{01} = \frac{C(S_{lo}) + C(S_{hi})}{L}$ |
| RIGHT error count | $f_{02} = C(S_{hi})$ |
| LEFT error count | $f_{03} = C(S_{lo})$ |
| OVERALL error count | $f_{04} = C(S_{ov})$ |
| RIGHT/OVERALL | $f_{05} = \frac{C(S_{hi})}{L}$ |
| LEFT/OVERALL | $f_{06} = \frac{C(S_{lo})}{L}$ |
| (RIGHT - LEFT)/overall | $f_{07} = \frac{C(S_{hi}) - C(S_{lo})}{L}$ |
| RIGHT anomaly signal | $f_{08} = Z(S_{hi})$ |
| LEFT anomaly signal | $f_{09} = Z(S_{lo})$ |
| OVERALL anomaly signal | $f_{10} = Z(S_{ov})$ |
| RIGHT norm signal | $f_{11} = \frac{Z(S_{hi})}{L}$ |
| LEFT norm signal | $f_{12} = \frac{Z(S_{lo})}{L}$ |
| OVERALL norm signal | $f_{13} = \frac{Z(S_{ov})}{L}$ |
| ANOMALY error | $f_{14} = Z(S_{lo}) + Z(S_{hi})$ |
| WITHIN anomaly error | $f_{15} = Z(S_{mi})$ |
| OVERALL error | $f_{16} = Z(S_{ov})$ |
| ANOMALY/WITHIN ratio | $f_{17} = \frac{Z(S_{lo}) + Z(S_{hi})}{Z(S_{mi})}$ |
| ANOMALY/num errors | $f_{18} = \frac{Z(S_{lo}) + Z(S_{hi})}{C(S_{lo}) + C(S_{hi})}$ |
| WITHIN/(n - num errors) | $f_{19} = \frac{Z(S_{mi})}{C(S_{mi}) + 1}$ |
| normalized WITHIN area | $f_{20} = \frac{Z(S_{mi})}{L}$ |
| WITHIN/OVERALL ratio | $f_{21} = \frac{Z(S_{mi})}{Z(S_{ov})}$ |
| norm. OVERALL error ratio | $f_{22} = \frac{Z(S_{ov})}{L}$ |
| average ANOMALY score | $f_{23} = \frac{Z(S_{lo}) + Z(S_{hi})}{C(S_{lo}) + C(S_{hi})}$ |
| average WITHIN score | $f_{24} = \frac{Z(S_{mi})}{C(S_{mi})}$ |
| average OVEREALL score | $f_{25} = \frac{Z(S_{ov})}{C(S_{ov})}$ |

Table 4: The basic anomaly detection building block features based on region-based aggregations.

| Feature name | Region-based specification |
|---|---|
| normative score: | $f_{26} = \text{score}(l_i)$ |
| normalized score: | $f_{27} = \frac{\text{score}(l_i)}{R}$ |
| flag:$e_{01}(l_i)$ | $f_{28} = e_{01}(l_i)$ |
| flag:$e_{02}(l_i)$ | $f_{29} = e_{02}(l_i)$ |
| flag:$e_{03}(l_i)$ | $f_{30} = e_{03}(l_i)$ |
| $\cdots$ | $f_{31} \cdots f_{50}$ |

Table 5: Twenty five additional features are used to identify whether a given feature value $f_r(l_i)$ from Table 4 represent extrema events with respect to the overall distribution and percentiles of feature values obtained for normative natural samples.

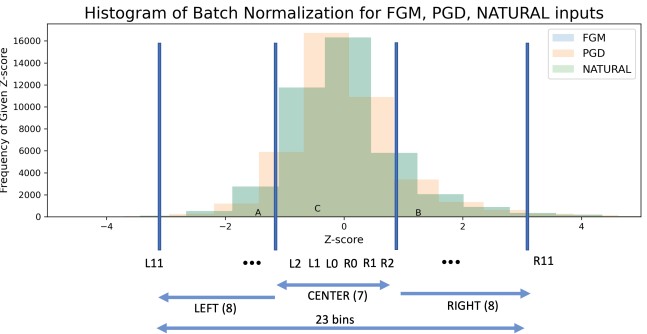

Figure 7: To compare the observed distribution $z_i$ values against the expected distribution of $z_i^N$ values histograms are used. We use 23 bins and construct features that measure discrepancies between observed and expected bincounts as well as symmetry between selected regions $R_x$ and $R_y$ of the compated distributions.

mative values and values outside will be flagged as outliers. After computing these $R$ boolean flags for every layer input values $l_i$, we compute a normative score as simply:

$$\text{score}(l_i) = \sum_{\forall r \in [1, R]} e_r(l_i).$$

We also add a normalized score, which divides the normative score by the number of region-based features $R$. A total of 25 extra features (see Table 5) are thus generated to flag unusually anomalous feature values.

### A.8. Basic and Comparative Histogram Features

In addition to the above $R$ region-based anomaly features, to train the neural network to understand fine-grained differences between the estimated $pdf$ of observed layer inputs $l_i$ and 10k randomly chosen normative layer inputs $l_i^N$ from the training set in $D_N$, histogram bincount features were

used over the standarized values $z_i$ of $l_i$, where $z_i$ is computed by subtracting the mean from $l_i$ and dividing by the standard deviation.

Until now, each of the features above represent features evaluated solely with respect to the observed layer inputs $l_i$. Next, we develop a set of comparative (logical/boolean, differential, ratio)-based features, that seek to train the neural network to discern the observed distribution of $l_i$ values against the distribution of normative $l_T^N \in R^L$ layer input values — extracted by element-wise averaging across the $D$ normative natural samples in $D_N$:

$$l_T^N = \frac{\sum_{\forall i \in [1, D]} l_i^N}{D}.$$

That is, the average $l_T^N$ input layer values for natural samples at the preselected batch normalization layer is a vector of size $L$ which is the element-wise arithmetic mean of every $l_i^N$ from the normative set of $D_N$.

Given any vector of $l_i$ values of size $L = 40960$, a histogram of $B = 23$ bins is computed using the range $[-3, 3]$, resulting in intervals of size $q = \frac{6\hat{\sigma}}{B}$. The resulting $B = 23$ bincounts $(h_{01}(l_i^*) \cdots h_B(l_i^*))$ represent 23 additional features.[3]

Moreover, just as we generate $B = 23$ histogram bincount features $(h_{01}(l_i) \cdots h_B(l_i))$ for observed layer input values $l_i$, we also compute the histogram bincounts $(h_{01}(l_i^N) \cdots h_B(l_i^N))$ for the normative layer input values. Then, using these two sets of $B = 23$ histogram bincounts (i.e., observed bincounts against normative bincounts), we compute features that magnify dissimilarity (specifically, square difference as well as relative error) between observed bincounts for a sample and the reference normative bincounts from natural samples. In all, the following three sets of $B = 23$ additional features are added:

- ($B = 23$ features $f_{51} \cdots f_{74}$:) the OBSERVED bincounts for observed layer values $l_i$: $h_{01}(l_i) \cdots h_B(l_i)$,

- ($B = 23$ features $f_{75} \cdots f_{98}$ :) the SQUARE DIFFERENCE between observed and normative bincounts: $(h_{01}(l_i) - h_{01}(l_T^N))^2 \cdots (h_B(l_i) - h_B(l_T^N))^2$,

- ($B = 23$ features $f_{99} \cdots f_{122}$:) the RELATIVE ERROR between observed and normative bincounts: $\frac{h_{01}(l_i) - h_{01}(l_T^N)}{h_{01}(l_T^N)} \cdots \frac{h_B(l_i) - h_B(l_T^N)}{h_B(l_T^N)}$

- and (3 features $f_{123} \cdots f_{125}$:) the sum, mean, and variance of the RELATIVE ERROR above.

The fine-grain targeting scope of our binning features is illustrated in Fig. 7 by juxtaposing along the three previously discussed coarse-grain aggregation regions.

### A.9. Comparative A/B Probability Test Features

A couple of A/B tests are used to compare the size $L = 40960$ populations of observed $l_i$ layer values from normative layer values. We examine whether observed values are normally distributed, whether observed values have a similar mean to that of normative values, whether channel means of observed values are similar to the means of channels in normative values, whether observed values have the same distribution as normative values, and whether observed values have a similar variance to that of normative values. These features are described in Table 6.

### A.10. Wasserstein Distance Features

Given the bin-counts from Section A.8, the earth moving distances (i.e., Wasserstein distances) compute how much

| Feature name | Feature description |
|---|---|
| normally distributed? | $f_{124} = \text{KolmSmirnov-test}(l_i, l_T^N)$ |
| same channel means? | $f_{125} = \text{mannwhitneyu-test}(l_i, l_T^N)$ |
| same pop. means? | $f_{126} = \text{t-test}(l_i, l_T^N)$ |
| same distribution? | $f_{127} = \text{ks-test}(l_i, l_T^N)$ |
| same variance? | $f_{128} = \text{bartlett-test}(l_i, l_T^N)$ |

Table 6: P-values resulting from the application of several statistical tests are used as features. These tests compare a random small subsample of the population of 40960 observed layer input values against a random small subsample of the population of 40960 normative layer input values.

work is needed to make look alike the skyline of a histogram $H_1$ to the skyline of a reference histogram $H_0$. Here, $H_1$ represents the histogram for the distribution of observed layer input values for a given sample and $H_0$ represents the histogram for the distribution of normative layer input values – that is, extracted from natural samples. Our Wasserstein histogram bin-count comparison features, built using features $f_{99}$ through $f_{122}$, compare:

- ($H_1$) Left Tail from Observed Distribution for the layer inputs of a given sample against ($H_0$) Left Tail of the distribution across all natural samples.

- ($H_1$) Right Tail from Observed Distribution for the layer inputs of a given sample against ($H_0$) Right Tail of the distribution across all natural samples.

- ($H_1$) Center Tail from Observed Distribution for the layer inputs of a given sample against ($H_0$) Center Tail of the distribution across all natural samples.

This hypothesis tests whether the shape of the *pdf* carries anomaly detection value.

### A.11. Dimensionality Reduction Features

We generate PCA(ndim=2) and LDA(ndim=2) features to test whether separability features improve classification accuracy. Fig. 8 shows a xy-plot of AFVs using the first two LDA features. It illustrates the spatial clustering patterns across samples of different attack methods and whether additional separatability features ought to be introduced and for which attacks.

### A.12. Nearest Neighbor Classification Support Features

Given a subsample of generated AFVs for both natural samples and samples perturbed by various attack methods, we trained a Radius Nearest Neighbor classifier (RNN). Then,

---

[3]These features provide a count of the number of standarized $z_i$ that fall into each of the $B = 23$ bins that span the interval $[-3, 3]$ of the normative range of standarized values.

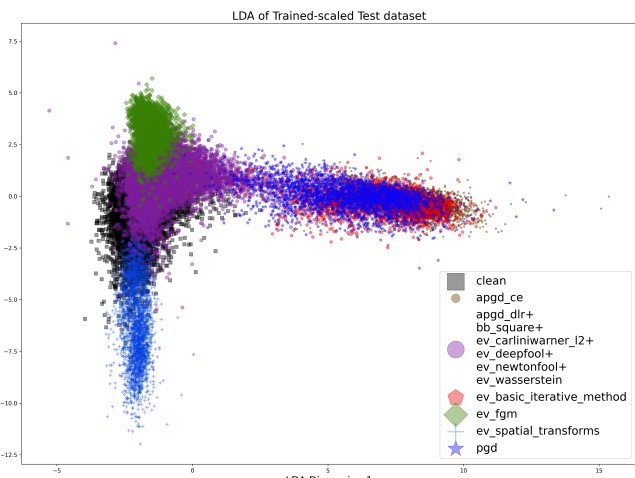

Figure 8: Plot of the first two standarized dimensions resulting from the LDA dimensionality reduction. Note how the $LDA$ first two dimensions provide some signal separation between natural samples (i.e., grey squares) and several attack classes? Can dimensionality reduction features help us discriminate anomalies between natural and attacked samples?

for any sample to be evaluated, we estimated dataset support for classification by means of the RNN's class voting probabilities and embedded them as features to the DeClaW Neural Network. The RNN classifier computes the number of samples of each class in the $(r = \epsilon)$-ball neighborhood of the anomaly feature vector to be classified. For CIFAR-10, we used $r = 3$ and $5K$ anomaly feature vectors.

## A.13. Additional Results

For the CIFAR-10 dataset, Fig. 9 reproduces Figure 6 at a larger scale as well as incorporating data for the CLEAN and APGD classes. Fig. 10 shows a xy-plot of TPR vs FPR values observed across hundreds of runs for CIFAR10 data. It shows that Declaw detection metrics consistently manifest desirable low FPR and high TPR – across the space of our grid parameter search.

Finally, DeClaW was preliminarily applied to the CIFAR-100 dataset. Without tuning, we achieved detection accuracy on CIFAR-100 (88% accuracy on adversarial data and 79% on clean data). As before, we were able to distinguish among many of the 11 different attacks on CIFAR-100 with F1-scores ranging from 74%-96%. Figs. 11- 12 show the corresponding detection and classification confusion matrices for the CIFAR-100 experiment.

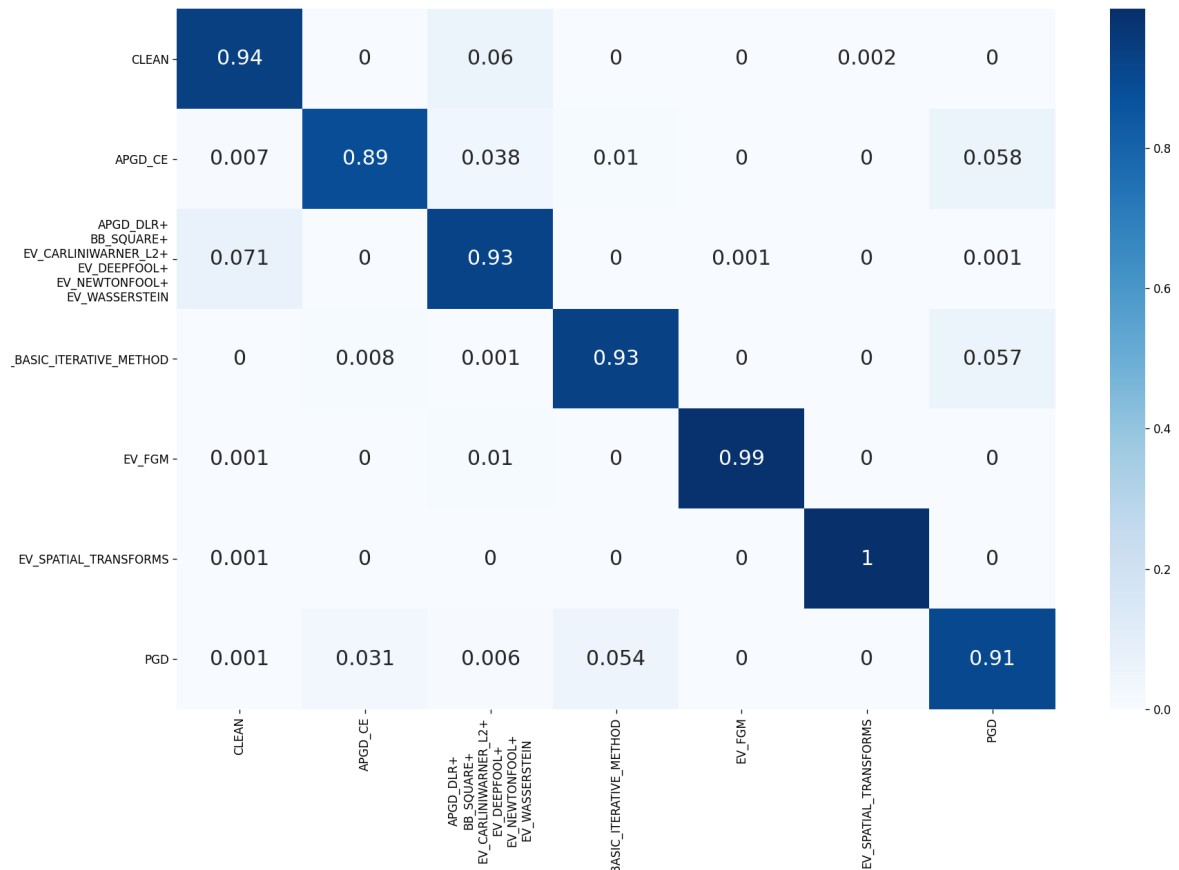

Figure 9: Confusion matrix for the classification results for the CIFAR-10 dataset produced by Declaw after the 12 classes above are clustered into six cluster groups based on large FPR and FNR.

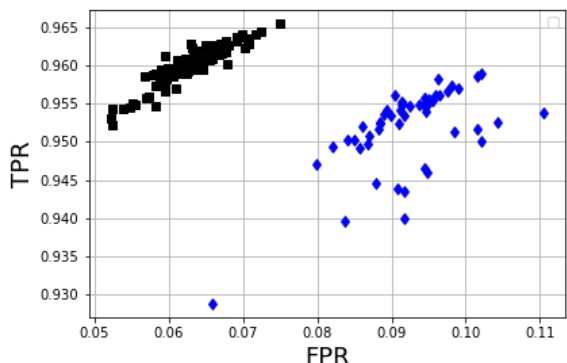

Figure 10: Plot of averaged TPR vs FPR values for the CIFAR-10 dataset across hundreds of different runs resulting from our grid parameter search. Blue diamonds correspond to results obtained with a learning rate of 0.01 and black squares correspond to values obtained using a learning rate of 1.00. Declaw detection consistently plots in the ROC sweetspot (i.e., low FPR, high TPR).

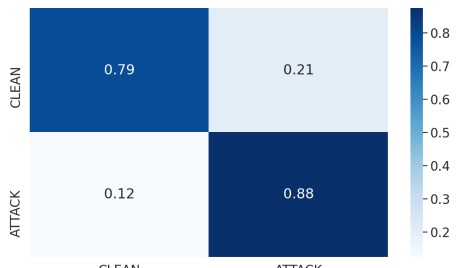

Figure 11: Confusion matrix for the detection of adversarial examples for the CIFAR-100 dataset where CLEAN bit is set only if no ATTACK class is detected and ATTACK bit is set otherwise.

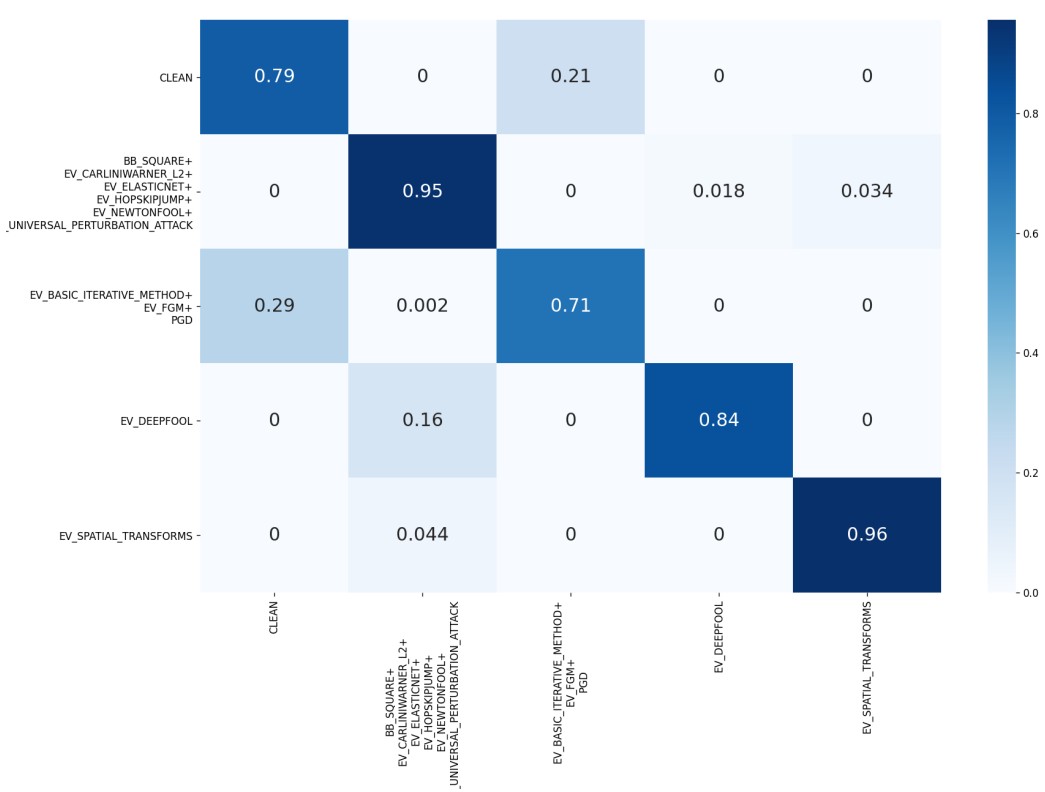

Figure 12: Confusion matrix for Declaw's classification accuracy for the CIFAR-100 dataset among attacks after attack methods are clustered into clusters based on large FPR and FNR.