# OpenReview forum: "Using Anomaly Feature Vectors for Detecting, Classifying and Warning of Outlier Adversarial Examples"
_ICML.cc/2021/Workshop/AML — ICML 2021 Workshop AML Poster_

### Official Review · Reviewer_5yos · 2021-06-20
**A method to classify clean image and adversarial examples crafted by different attack types**

**Rating:** Accept
**Confidence:** 5

**Review:**

This paper proposed a DNN-based attack-type classifier based on anomaly feature vector (AFV) on the CIFAR10 dataset. The authors augmented a pre-trained model with a sampling hook at a chosen layer to generate AFV of clean or adversarial examples. The AFVs from the perturbed inputs as well as natural inputs of different classes are used to train the classifier. The results show high accuracies both in detecting clean examples and classifying adversarial examples crafted by different attack algorithms.

Strengths:
1. This paper is well organized. The authors illustrated the statistical differences between different attack algorithms. They as well achieved high accuracies in classifying different attack types.

Weaknesses:
1. The meaning of classifying different attack types is not clear.
2. In this paper, the attack types are first clustered by a clustering algorithm where the attack types in one cluster are treated as one composite class, while the accuracy reported in the paper is based on these composite classes. Therefore, the result is unreliable because it depends on the additional clustering algorithm. For example, a naïve classifier could have a high classification accuracy if the classes are defined by a naïve clustering algorithm.

---

### Decision · Program_Chairs · 2021-06-21

**Decision:**

Accept (Poster)

**Comment:**

This paper proposed a DNN-based attack-type classifier based on anomaly feature vector (AFV). The paper can be improved by addressing the reviewer's comments.